# Evaluation of *Dosidicus gigas* Skin Extract as An Antioxidant and Preservative in Tuna Pâté

**DOI:** 10.3390/foods8120693

**Published:** 2019-12-17

**Authors:** Jesús Enrique Chan-Higuera, Josafat Marina Ezquerra-Brauer, Leontina Lipan, Marina Cano-Lamadrid, Roberta Rizzitano, Angel Antonio Carbonell-Barrachina

**Affiliations:** 1Grupo Calidad y Seguridad Alimentaria, CSA, Departamento de Tecnología Agroalimentaria, Escuela Politécnica Superior de Orihuela (EPSO), Universidad Miguel Hernández de Elche (UMH), Carretera de Beniel, km 3.2, 03312 Orihuela, Alicante, Spain; jeen.chhi@gmail.com (J.E.C.-H.); leontina.lipan@goumh.umh.es (L.L.); marina.cano.umh@gmail.com (M.C.-L.); robertarizzitano@alice.it (R.R.); 2Departamento de Investigación y Posgrado en Alimentos, Universidad de Sonora, Blvd, Luis Encinas y Rosales s/n, Col. Centro, C.P. 83000 Hermosillo, Sonora, Mexico

**Keywords:** antioxidant activity, antimicrobial activity, squid pigments, sensory analysis

## Abstract

A strategy for food preservation, based on a methanol–HCl squid skin extract (*Dosidicus gigas*) (JSSE), was evaluated at two concentrations in yellowfin tuna fish pâtés, which were stored at 4 and 8 °C for 20 day. The JSSE was characterized by determining its antioxidant and mutagenic activities. A yellowfin tuna pâté was elaborated, with and without the addition of the JSSE. An affective sensory analysis was performed to establish consumers’ preferences. During a 20-day storage period, the water activity (a_w_), pH, color difference (ΔE*_ab_), microbiological analysis, lipid oxidation and sensory quality attributes were evaluated, and the results were compared with the results of the butylated hydroxyanisole (BHA) and control treatments. The JSSE showed antioxidant activity against the 2,2-diphenyl-1-picrylhydrazyl (DPPH^●+^) and 2,2′-azino-bis (3-ethylbenzothiazoline-6-sulphonic acid) (ABTS^●^) radicals and did not induce mutation, according to the Ames’ *Salmonella* test, nor chromosomal abnormalities, according to the onion root-tip cell assay. The consumer analysis demonstrated a higher preference for the pâté with the added JSSE in seven out of the eight evaluated attributes. During storage, the JSSE neither had an impact on a_w_ nor pH, maintained lower ΔE*_ab_ values, inhibited the microbial activity and lipid oxidation (unlike the control pâté), and preserved the sensory quality attributes, unlike the BHA and control treatments. This study showed that the JSSE has biologically active pigments that can act as antioxidants and antimicrobials in yellowfin tuna fish pâtés.

## 1. Introduction

Fatty fish-based products, such as fish pâté, are food items of great economic and nutritional importance because of their composition and health benefits [1]. Nonetheless, during their processing and storage, microbial growth and lipid peroxidation in these fatty fish-based products lead to sensory and nutritional quality losses. Microorganisms can directly affect food through the development of undesirable flavors, odors and colors, rendering products unsuitable for human consumption [2]. Oxidative reactions cause the degradation of nucleic acids, proteins, lipids and pigments. Lipid peroxidation is responsible for detrimental changes in sensory attributes, as well as the production of toxic compounds [3].

Fish pâté is widely regarded as a product with an important gastronomic tradition and a high nutritional value, as well as appreciated sensory attributes. Pâté is more vulnerable to microbial growth and peroxidation due to the disruption of its cellular components, a process which is induced by the grinding process [4]. Minced muscle has a lower shelf life because of the increase in nutrient availability and moisture migration, both of which enable microbial development. Moreover, exposed fatty acids can react with pro-oxidant molecules and interact with oxygen, light and metals, all of which promote fatty acid oxidation [5]. In the formulation of fish pâté, the use of fatty fish filets (specifically from yellowfin tuna, *Thunnus albacares*) is preferred because of their palatable characteristics.

Since domestic refrigeration conditions range from 6 to 11 °C, instead of the recommended temperature of 4 °C, the use of additives in refrigerated products is necessary to prevent microbial growth and lipid oxidation [6]. Even if synthetic antioxidants and antimicrobials are highly effective, there is uncertainty about their negative impact on human health. Replacing synthetic additives with natural compounds that can exert the same function satisfies the popular demand for clean labels and contributes to the creation of safer products [7].

Biologically active compounds have been identified and characterized from different natural sources: meat, fish and vegetable peptides, phenolic compounds from spices and herbs, vegetable pigments, etc. [8]. One type of biologically active compound is represented by ommochromes, which are pigments found in the eyes of invertebrates, like crustaceans and arthropods, as well as the skin of mollusks. Chemically, ommochromes have a basic structure that consists of a phenoxazine ring with different substituents. They are considered to be tryptophan-derived metabolites that come from the kynurenine pathway [9]. Their functional groups, which vary from molecule to molecule, give ommochromes a distinctive reddish-to-violet coloration, as well as antiradical in silico activity [10]. These compounds act as antioxidants against UV radiation [11]. Studies of jumbo squid (*Dosidicus gigas*) skin have demonstrated its antioxidant activity, applied in fish oil at different storage temperatures [12], as well as its antimicrobial activity during the chilled storage of fresh mackerel and hake [13,14]. All these results suggest that ommochromes can be used as additives in the food industry; however, there is still no information regarding the use of jumbo squid skin extracts (JSSEs) in food matrixes, like fish pâté, which tend to be more complex and more susceptible to oxidation reactions.

In this work, the main objectives were to evaluate the biological activity of giant squid skin extracts in the oxidative stability of lipids, the microbial growth of mesophiles and psychrophiles, and the loss of sensory quality of yellowfin tuna pâtés, which were stored at two temperatures (4 and 8 °C). In order to carry out the addition of the extract to the tuna pâtés, the in vitro antioxidant activity and the mutagenic activity of the extracts were also evaluated.

## 2. Materials and Methods

### 2.1. Jumbo Squid Skin Extraction and Preliminary Analysis

JSSEs were obtained by using acidified methanol (1% HCl). A preliminary study was conducted to determine the effect of the skin:solvent ratio, sonication time, and extraction temperature on the recovery of pigments with antioxidant activity from squid skin. It was found that 20 g mL^−1^ of the JSSE and sonication for 5 min at 25 °C provided the maximum recovery of pigments with antioxidant activity. The extracts were centrifuged (Model Biofuge Stratos, Thermo Scientific, Germany) at 10,000× *g* for 15 min before the methanol was removed by using a rotary evaporator (R-100, Büchi, Switzerland). The solvent was further evaporated by using nitrogen gas. Prior to further analysis, the dry extracts were stored in an inert nitrogen atmosphere at −80 °C.

#### 2.1.1. In Vitro Antioxidant Activity

The free radical scavenging activity of the JSSE was determined by using the 2,2-diphenyl-1-picrylhydrazyl (DPPH) radical scavenging and the 2,2′-azino-bis (3-ethylbenzothiazoline-6-sulphonic acid) (ABTS) assay, which has previously been described [15,16]. Samples of 1 mg per mL JSSE were dissolved in aqueous methanol (80%). The inhibition percentage of the extracts (IP) was calculated for both methods and expressed per mg of the extract by using Equation (1):
(1)$$\mathrm{IP}=100-\left(\frac{\left(A_{sample}\right)\left(100\right)}{A_{blank}}\right)$$
where *A_sample_* is the absorbance of the aqueous methanol and *A_blan_* is the absorbance of the extract solution.

#### 2.1.2. Salmonella Mutagenic Assay

The mutagenic activity was tested by using the *Salmonella typhimurium* tester strains, TA98 and TA100, which were purchased from Molecular Toxicology Inc. (MolTox; Annapolis MD, USA) with and without metabolic activation (presence of an S9 enzyme mix, MolTox). Four doses of the JSSE were evaluated (0.1, 0.5, 1 and 5 mg mL^−1^). All of them were diluted in a 0.2 M phosphate buffer (pH 7.4). The concentrations were selected based on a preliminary toxicity test. The plates were incubated at 37 °C for 48 h, and the revertant colonies were manually counted. All experiments were analyzed in triplicate. A sample was considered mutagenic when a dose–response relationship was detected, and a two-fold increase in the number of spontaneous mutants (MI ≥ 2) was observed with at least one concentration [17]. The standard mutagens that were used as positive controls in the experiments without the S9 mix were hydrogen peroxide (340 µg plate^−1^) for TA98 and sodium azide (1.25 µg plate^−1^) for TA100. Aflatoxin B_1_ (0.5 µg plate^−1^) was used with TA98 and TA100 in the experiments with metabolic activation. A phosphate buffer served as the negative (solvent) control.

#### 2.1.3. Onion Root-Tip Clastogenicity Test

Healthy young onion bulbs, grown in the absence of herbicides, pesticides or fungicides, were used in this study. The promotion of root development was performed by placing the bulbs in darkness and partially submerging them in water. When the roots were 2 cm long, the onions were transferred to Petri dishes with 30 mL of the JSSE extracts (0.1, 0.5, 1 and 5 mg each). A positive control with a sodium azide solution (10 ng mL^−1^) and a negative control (water) were also analyzed. After 24 h, the roots were fixed in 3:1 (*v*/*v*) ethanol/glacial acetic acid, squashed and washed with distilled water, and stained with orcein for 2 h in the dark. All cells with alterations were counted [18]. During the genotoxicity assessment, the presence of mitotic cells with irregular chromosomes (e.g., micronuclei, disorganized chromosomal structure, lag and stick chromosomes) was recorded.

### 2.2. Pâté Elaboration

Tuna pâtés were produced in the food processing pilot plant at Miguel Hernández University (Alicante, Spain). Fillets of fresh yellowfin tuna fish were obtained from a local market in Orihuela (Alicante, Spain). Tuna fish fillets were chopped into small cubes and mixed for 15 min in a Vorwerk Thermomix food processor (Wuppertal, Germany). Tuna (40 g) was mixed with ice (30 g) and salt (1 g) for 5 min. Afterwards, sodium caseinate (4 g) and corn starch (4 g) were slowly added until a homogenous mix was formed. Olive oil (5 g) was smoothly added until the emulsion was formed. Finally, white wine vinegar (1 g) was added. All batches were cooked until the pâté core temperature reached 75 °C, and they were then cooled in an ice bath.

Based on preliminary studies, the JSSE was added separately in two batches at concentrations of 0.05% of pâté (P1) and 0.1% of pâté (P2). Butyl hydroxyanisole (BHA) (0.1% of pâté) was added to a third batch (BHA), representing a different treatment. The JSSE was mixed with vinegar before being mixed with the other ingredients. The remaining batch was used as a control sample (control).

#### Consumer Acceptance Panel

Pâté samples were taken from refrigerated storage and left at 20 °C for about 15 min. They were then analyzed by a panel of 70 consumers. Consumers were asked to state their preference in relation to eight attributes on a scale (1: extremely dislike it; 5: neither like nor dislike it; and 9: extremely like it). The consumers between 20 and 65 years old were students and staff members of the Department of Agro-Food Technology (Miguel Hernández University, Desamparados Campus). The sensory analyses were performed after the pâté was cooked.

### 2.3. Cooling Storage and Pâté Shelf Life

All samples were packed in polyethylene plastic bags and divided into vacuum sealed and non-vacuum sealed bags. They were then stored at 4 and 8 ± 1 °C under dark conditions for 20 day. Pâté samples were taken randomly at 0, 4, 8, 12, 16 and 20 day of storage. Analyses were performed on the sampling day. All determinations were performed in triplicate (*n* = 3).

#### 2.3.1. Physical-Chemical Parameters

The water activity was determined in polyethylene capsules with circa 5 g of pâté were placed in a Labmaster-aw instrument (Novasina, Lachen, Switzerland). Analyses were performed at 25 °C. The pH was evaluated by mixing 10 g of pâté with 100 mL of distilled water and then stirring the mixture in a magnetic plate. pH measurements were made by using a pH Basic 20 instrument (Crison, Barcelona, Spain). The color of the pâtés was measured by using a CR-400 Chromameter (Konica Minolta, Tokyo, Japan), and the *L**, *a** and *b** parameters were registered, as specified by the International Commission on Illumination. The total color difference over the time between samples was evaluated with the ΔE*_ab_ parameter, following Equation (2) [19]: (2)$$∆E_{ab}^{*}=\sqrt{{\left(L_{2}^{*}-L_{1}^{*}\right)}^{2}+{\left(a_{2}^{*}-a_{1}^{*}\right)}^{2}+\;{\left(b_{2}^{*}-b_{1}^{*}\right)}^{2}}$$

#### 2.3.2. Microbiological Analysis

Portions (20 g) of the samples (for each of the control and treatment groups, separately) from the yellowfin tuna pâté were aseptically taken and homogenized in sterilized stomacher bags (Seward, United Kingdom) with 90 mL of 0.1% peptone water. Serial tenfold dilutions were made in pre-sterilized tubes that contained 9 mL of peptone water. The sample preparation and plating were carried out under a laminar flux cabinet in sterile conditions. Triplicate samples were analyzed from each trial of pâté formulation. To estimate the total plate count, 1 mL of the dilutions was aseptically transferred to Petrifilm count plates (3M Corporation, Maplewood, MN, USA). Petrifilms were divided into two incubating conditions for the determination of the total aerobic mesophiles (37 °C) and psychrophiles (7 °C). Colonies were counted after 48 h for mesophiles and 72 h for psychrophiles by using a colony counter. The average count was multiplied by the dilution factor and expressed as the colony-forming units per gram (CFU) of the sample.

#### 2.3.3. Lipid Oxidation

Conjugated dienes (CD) and trienes (CT) were measured by mixing a 0.5 g sample of pâté with 5 mL of distilled water and vortexed at 1500 rpm for 1 min. An aliquot of 0.5 mL was mixed with 2.5 mL of hexane:isopropanol (3:2 *v*/*v*) and centrifuged at 2000 rpm for 3 min. The supernatant was recovered and placed in quartz cuvettes, and measurements of CD and CT were conducted at 232 and 268 nm, respectively. The concentration of dienes and trienes was determined by using the molar extinction coefficient, specific for both types of compounds [20].

The quantification of hydroperoxides, evaluated as secondary peroxidation products, was carried out according to previous reports [21], with some modifications. Lipid extraction was conducted by using hexane:isopropanol. The extract (1 mL) was mixed with 10 mL of chloroform:methanol (7:3 *v*/*v*) and vortexed for 30 s at 2000 rpm. Later, 50 µL of 30% ammonium thiocyanate and 50 µL of 20 mM iron (II) chloride were added and centrifuged at 2500 rpm for 10 min. The absorbance of the supernatant was obtained at 480 nm, and the concentration of peroxides was expressed as the equivalent nmol of the cumene hydroperoxide per g of pâté.

The end-products of the peroxidation were determined by using a thiobarbituric acid assay (TBARS) [22]. A sample of 0.5 g was mixed with 2.5 mL of thiobarbituric acid reagent (3.75 g of TBA, 150 g of trichloroacetic acid in 1 L of 0.25 N HCl) and heated in a water bath at 96 °C for 10 min. The test tubes were cooled down in an ice bath and centrifuged at 2500 rpm for 10 min. The supernatant was recovered, and its absorbance was recorded at 532 nm. The standard curve was prepared with 1,1,3,3-tetramethoxypropane, and the results were expressed as the mg of malondialdehyde per kg of the pâté.

The antioxidant efficacy of the JSSE in the samples was assessed with an oxidation index at both storage temperatures (4 and 8 °C). The percentage of oxidation inhibition (OI (%)) was calculated for peroxide value (PV) and TBARS by using Equation (3) [23]:(3)$$\mathrm{OI}\;\left(\%\right)\;=\left(\frac{c-s}{c}\right)\times \;100$$
where *c* is the value obtained from the control on the day of the highest obtained value and *s* is the value for each JSSE condition on the same day.

#### 2.3.4. Sensory Quality

To evaluate the sensory quality of the samples during storage, a panel of seven qualified and experienced panelists in the field of fish products was appointed, and they used the sensory attribute descriptors of the coded samples. They performed the analysis in a standard sensory laboratory (under white light, 25 °C, and 50–55% relative humidity). The same panel evaluated all the samples. The analysis was performed by scoring sensory properties by assigning five categories: highest quality (E), good quality (A), fair quality (B), poor quality (C), and unacceptable quality (D). The sensory descriptors of the pâté were odor, the presence of off-colors, the presence of slime, firmness, and emulsion stability. The attribute descriptors are shown in Table 1.

### 2.4. Statistical Analysis

Descriptive statistics were used to present the antioxidant assays (*n* = 3). The data obtained from the consumer acceptance study were evaluated through a one-way analysis of variance (ANOVA) in order to assess the significance of the JSSE addition on all of the studied characteristics. Data from the color and lipid oxidation of the pâté samples were evaluated through a two-way analysis of variance (ANOVA). The data obtained from the Ames’ test and the microbiological and chemical analyses (*n* = 3) were subjected to the ANOVA analysis to establish differences that had resulted from the effects of the JSSE addition. A comparison of the means was performed by using the Tukey’s honestly significant test (*p* < 0.05). The data obtained from the sensory evaluation and the onion root tip assay were analyzed by using the non-parametric Kruskal–Wallis test. In all cases, the analyses were carried out by using SAS software (SAS Institute, Inc. JMP 5.0.1, Cary, NC, USA), and differences among treatments were considered significant with a confidence interval at *p* < 0.05 in all cases.

## 3. Results

### 3.1. Preliminary Analysis of Jumbo Squid Skin Extracts

The obtained extract showed a reddish-violet color, and the recovery yield (expressed as g dry JSSE 100 g^−1^ skin) was 0.65%. The antioxidant compounds present in the 1 mg mL^−1^ of the JSSE generated an inhibition percentage of 68 against the DPPH* radical and 79 in the ABTS^●^* test.

The mutagenicity of the JSSE against both *Salmonella* tester strains was considered negative, because the extracts did not double the number of spontaneous revertant colonies, counted per plate (Figure 1).

The JSSE clastogenicity test showed that the obtained results, which were determined by the onion root-tip assay, could be considered as non-genotoxic. The percentage of cells with damaged chromosomes was lower than 50%, as compared to the sodium azide root (Table 2). The method was validated by the percentage of observed cells (over 150) and mitotic cells (over 100).

### 3.2. Consumer Acceptance

The evaluation of consumer acceptance showed that the JSSE addition in the tuna pâtés improved their sensory attributes (Table 3).

### 3.3. Assessment of Quality Evolution During Fish Pâté Cooling Storage

#### 3.3.1. Physical-Chemical Parameters

The water activity of all of the tuna pâtés ranged from 0.93 to 0.95 during all samplings. When compared with the control treatments, it could be seen that the JSSE addition did not directly affect the proportion of free water in the samples, (*p* > 0.05). The results concerning the pH levels showed no differences among the samples (*p* > 0.05). Nevertheless, a significant pH drop, from about 6.4 to 5.7, was detected at storage day four for all pâté treatments.

The color of the samples became more intense when the amount of the JSSE added to the tuna pâté increased (Table 4). The changes in the ΔE*_ab_ values during the storage days were statistically significant, as shown in Table 5. In the period of 12–20 day at 4 °C, the mean value changes were greater than five units in the control and BHA treatments, whereas at 8 °C, they were less than five units on the eighth day in the JSSE and BHA pâtés.

#### 3.3.2. Microbiological Analysis

The changes with time of the aerobic mesophiles and psychrotrophs in the four types of pâté are shown in Table 6. In the period of 12–20 day, the mean of the aerobic concentrations of mesophiles was greater than 250 logarithmic units in the control and BHA, whereas the concentrations of psychotrophs were less than 250 logarithmic units on the 8th day and 4th day at 4 and 8 °C, respectively, in the JSSE-pâté.

#### 3.3.3. Lipid Oxidation Index

Comparative studies on the level of the primary oxidation products, conjugated dienes (CD), and conjugated trienes (CT) are shown in Table 7. At both storage temperatures (4 and 8 °C), the concentration of CD and CT increased from day 16 to 20. A definitive trend due to the presence of the JSSE in tuna pâté could not be demonstrated.

The maximum of the hydroperoxide values at 4 °C was detected on day four for all treatments (Figure 2), whereas at 8 °C, the maximum was only observed in the control, BHA and P1. At both temperatures, the sample treated with the higher concentration of the JSSE (P2) exhibited period values under the 0.5 mM equivalents of PV, indicating an inhibitory effect (*p* < 0.05). On day four of the experiment, at both temperatures, the following hydroperoxide formation was detected: control = BHA = P1 < P2.

The end products of the lipid peroxidation found in the pâté samples were evaluated by using the TBARS method (Figure 3). As a general trend, this inhibitory effect (*p* < 0.05) was higher in P2 throughout the entire experiment at both storage temperatures, and it was also higher (*p* < 0.05) than the control pâtés. As expected, BHA exerted the highest inhibitive effect, when compared to all the other treatments (*p* < 0.05).

The inhibition percentage of the JSSE in the pâté lipid oxidation was calculated for PV and TBARS at 4 and 8 °C based on the control condition scores obtained. The percentage of oxidation inhibition, measured by using the PV index for the BHA and JSSE samples, indicated that the pâté with 0.1% of the JSSE (P2) was the sample that showed the highest inhibition at 4 °C. The order of oxidation inhibition values in the peroxide formation at 4 °C was P2 (41.5%) > BHA (14.9%) > P1 (6.2%), and at 8 °C, it was P2 (57.8%) > BHA (9.0%) > P1 (2.5%). The TBARS content inhibition at 4 °C was BHA (76%) > P1 (33.9%) > P2 (17.9%), and at 8 °C, it was BHA (73.9%) > P2 (45%) > P1 (25.5%).

#### 3.3.4. Sensory Evaluation

The evolution of the sensory qualities of the pâtés is depicted in Table 8. The initial pâté types were found to have the highest quality (E score). After eight days of storage at 4 °C, samples corresponding to the P1 (0.05% JSSE) treatments showed a good quality and were considered to be in the A category. Remarkably, the pâté types belonging to the control, BHA and P2 (0.1%) treatments showed a fair or poor quality, as compared with their P1 counterparts at this storage time. Meanwhile, at 8 °C, until day 12, the sensory quality of the P1 treatment was considered good, whereas the control, BHA and P2 had a fair or poor quality at this period of time.

## 4. Discussion

The main objective of the first part of the study was to characterize the obtained jumbo squid pigment extracts, mainly their antioxidant activity and potential toxic effect. The antioxidant activity of the extracts was measured in terms of their DPPH* and ABTS^•^* radical scavenging activities, whereas the mutagenicity test with *Salmonella* strains and the onion root-tip clastogenicity tests were used to evaluate the potential toxic effect of a sample on the genetic material in prokaryotic and eukaryotic cells. The antioxidant activity of the JSSE was considered effective because it was higher than the results reported for DPPH* (30%) and for ABTS^•^* (50%) [24]. The *Salmonella* test indicated that the mutagenicity was considered negative when the number of revertant colonies counted per plate did not double the number of spontaneous revertants [25]. Based on this test, the mutagenicity of the extracts was considered negative. Similarly, for plant-based products and purple natural pigments, no mutagenic effect on the same *Salmonella* strains was detected [26,27]. Concerning the onion root-tip test, there was genotoxic activity when the percentage of cells with damaged chromosomes was over 50%, as compared to the sodium azide roots. None of the JSSE treatments evaluated (0.1, 0.5, 1 and 5 mg mL^−1^) could be considered as clastogenic, according to the results of this study.

The objective of the second part of the study was to establish if the addition of the JSSE would affect the consumer acceptability of the tuna pâtés. Consumers’ sensory results demonstrated that the JSSE-treated samples (0.05 and 0.1%) had significantly higher scores (*p* < 0.05), as compared to the control and BHA samples, in seven out of the eight evaluated attributes (color, odor, tuna flavor, spreadability, cohesiveness, aftertaste, and overall acceptance). Therefore, the evaluated JSSE concentrations were accepted to a higher degree, as compared to the control samples.

The third phase of the study was designed to establish the effect of the JSSE on the tuna pâtés’ storage life at domestic refrigeration conditions (8 °C) and recommended temperature (4 °C). Concerning the physical–chemical results, color was the only parameter of the pâtés that was affected by the JSSE. The decrease in the pH values on day four may have been due to the microbial development, resulting in the formation and accumulation of acidic compounds, such as lactic acid, among other acidic metabolites [28]. Regarding color, values greater than 5 are considered to be perceptible to the human eye [18]. At the storage temperature of 4 °C for the control treatment, the values of ∆E were greater than 5 from day four onwards, and for the BHT treatment, they were greater than 5 from day eight. JSSE-pâtés (P1 and P2) maintained ΔE*_ab_ values of less than 5 until day 12 of storage, whereas at 8 °C, samples with the JSSE and BHT maintained their color up until day eight of the experiment. Therefore, the results demonstrated that samples with the JSSE maintained their color for at least eight days, depending on the storage temperature.

The inclusion of the JSSE in the pâté caused a better microbial growth control compared to the control and BHA treatments. The maximum allowed levels of microorganisms established for meat products, cured and cooked meat products, and emulsified and cooked cured products are 250 CFU g^−1^ [29]. Species-specific spoilage psychrotrophs bacteria, including members of the genera *Pseudomonas, Shewanella, Acinetobacter, Moraxella* or *Flavobacterium*, are considered to be in the psychrotrophic group. The results of this experiment demonstrated a bacteriostatic effect on tuna pâté due to the inclusion of a squid skin extract (JSSE) in the formulation at both 4 and 8 °C storage temperatures and at both addition concentrations. Previous studies have also evaluated the preservative effect of acetic acid–ethanol extracts of jumbo squid skin in chilled fish [13,14]. In those studies, an antimicrobial effect (lower aerobic mesophiles and psychrotrophs counts) was observed during the chilled storage of fresh fish. This behavior may be due to the electron donor capacity of the ommochromes [10], inducing an imbalance in the metabolic pathways in microorganisms. However, further research is needed to establish the possible mechanism of the compounds present in the extract.

Regarding lipid oxidation, the addition of the JSSE did not improve or promote the formation of primary peroxidation products. At both storage temperatures (4 and 8 °C), the concentration of CD and CT increased from day 16 and 20. This behavior indicated the isomerization of new fatty acids in the product, as mediated by free radicals. CD and CT did not have enough stability to maintain their structure, and they tended to associate with other components of the food or to decompose. Similar results have been reported in products such as vegetable oils [30]. Through the oxidation process, the quantity of the different molecules that were generated in these phases sequentially increased and then decreased over time. Peroxide formation was measured as an estimation of the propagation phase of rancidity. At this stage, molecular oxygen compounded with unsaturated fatty acids, thus generating hydroperoxides and free radicals, which, at the same time, further reacted with lipid molecules to develop other reactive chemical species [31]. It was established that the PV maximum tolerance value should be a 0.5 mM cumene hydroperoxide equivalent [21]. Thus, the inclusion of 0.1% JSSE in the tuna pâté did lead to an inhibition of PV at both temperatures.

The increase in TBARS denoted an aldehyde compound formation. The decrease in this indicator implied that the volatile aldehydes were transformed into other compounds [31], as was detected for the control and the P1 and P2 treatments at 4 °C and in the control and P1 treatments at 8 °C. The maximum allowed value for the TBARS values is a 2 mg malondialdehyde (MDA) kg^−1^ sample [22]. Control samples showed the maximum value at day 16 of storage at 4 °C. An inhibitory effect on the TBARS formation was observed for the two JSSE concentrations tested throughout the entire storage period.

Peroxides are the most reactive compounds that are generated during lipid oxidation. These findings strongly suggested that the JSSE acts as a secondary antioxidant, interacting directly with the radicals derived from fatty acids. Similar results have been reported in other products, such as pork sausages [32], chicken fillets [33], and salmon paste [34]. These results proved the protective effect of the JSSE against lipid oxidation in the pâté samples.

During storage, a gradual quality loss was perceived by the trained panel in the different sensory descriptors for all treatments. However, after day eight of storage at 4 °C and after day 12 at 8 °C, samples with 0.05% JSSE showed a good quality. The previous results were unexpected because the chemical and microbiological assays suggested that the P2 sample (0.1% of JSSE) had better results than the other treatments. P2 had the same overall sensory scores compared to the control, whereas BHA had lower scores throughout the experiment. This could be attributed to the development and increased production of tuna odors, as well as those associated to the addition of BHA and a higher squid skin extracts during storage time. This behavior was similar to that shown in other studies [35], which reported an increased quality loss up until the end of the analysis when studying minced chicken products. Other reports described a sensory quality improvement and increased shelf life in chilled mackerel and hake as a result of including squid skin extracts in the icing medium [12,13].

## 5. Conclusions

This work demonstrated that pigments extracted from squid skin with antioxidant and antimicrobial activity, as well as no mutagenic or clastogenic effects, can be used in the formulation of tuna fish pâté. When used at 0.05% (*w*/*w*), such an extract provided significant antioxidant and antimicrobial effects on pâté samples during storage at two different temperatures (4 and 8 °C) for 12 days. The loss of sensory quality also confirmed that the samples with 0.05% of the JSSE (P1) extract exhibited an extended shelf life when compared to other pâtés, being acceptable even after 12 day of chilled storage. These results provide a way in which the jumbo squid by-product can be used as a source of biologically active compounds.

## Figures and Tables

**Figure 1 foods-08-00693-f001:**
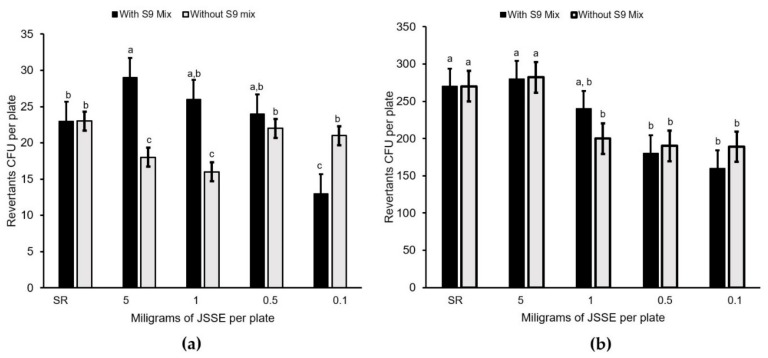
Mutagenic effect of the giant squid skin extract on (**a**) the TA98 and (**b**) TA100 strains with and without metabolic activation (S9 enzyme mix). Mean values are of three replicates (*n* = 3), and standard deviations are indicated by bars. Values with different letters are significantly different (*p* < 0.05). SR = spontaneous revertants per plate.

**Figure 2 foods-08-00693-f002:**
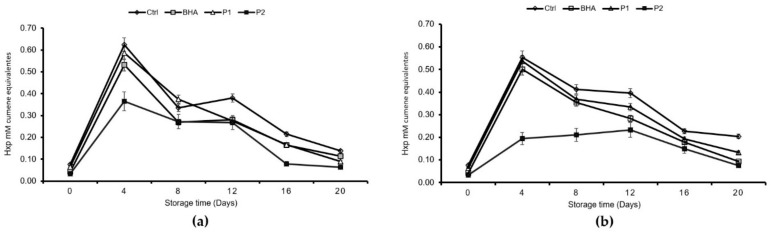
Hydroperoxide formation (Hxp mM cumene equivalents) * in the different processed pâtés ** during storage at (**a**) 4 °C and (**b**) 8 °C. * Mean values are of three replicates (*n* = 3), and standard deviations are indicated by bars. ** Abbreviations of BHA and jumbo squid extracts (P1 and P2) are the same as those expressed in Table 3.

**Figure 3 foods-08-00693-f003:**
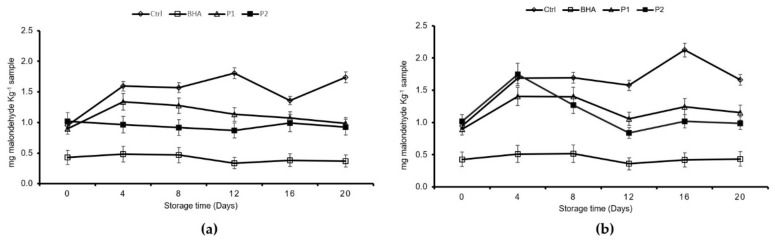
Formation of thiobarbituric acid reactive substances * in the different processed pâtés ** during storage at (**a**) 4 °C and (**b**) 8 °C. * Mean values are of three replicates (*n* = 3), and standard deviations are indicated by bars. ** Abbreviations of BHA and jumbo squid extracts (P1 and P2) are the same as those expressed in Table 3.

**Table 1 foods-08-00693-t001:** Scale used in the evaluation of the sensory quality of yellowfin tuna pâtés.

Descriptor Attribute.	Excellent (E)	Good (A)	Fair (B)	Poor (C)	Unacceptable (D)
Odor	Strong seaweed and shellfish	Weak seaweed and shellfish	Imperceptible seaweed and shellfish smell	Initial ammonia and putrid odor	Ammonia and putrid odor
Off-colors	Bright coloration; complete absence of decolorated areas	Dim coloration, small (<5 mm) decolorated areas	Dim coloration, larger (<10 mm) decolorated areas	Brownish coloration; presence of small greenish spots	Brown coloration throughout; greenish spots throughout
Slime	Complete absence of slime in the package and the sample	Droplets of transparent slime in the bags, located only in the corners	Slime present at the bottom of the bag; sample with slime droplets	Whitish slime present throughout package and sample	Opaque slime present throughout package and sample
Firmness	Elastic and spreadable samples; cohesive and moist particles	Firm consistency; less spreadable sample	Presence of thin cracks in the samples; complete loss of elasticity	Presence of wide cracks; separated pâté particles	Complete loss of water, important shape changes throughout
Emulsion stability	Homogeneous appearance throughout; no sign of oil separation	Homogeneous appearance in most of the sample; small droplets of oil	Heterogenous appearance in areas of the sample; droplets of water and oil at the surface	Heterogenous appearance in most of the sample; liquid condensed at the bottom of the package	Heterogeneous appearance throughout; water and oil separated at the bottom of the package

**Table 2 foods-08-00693-t002:** Clastogenic effect of the jumbo squid skin extracts (JSSE) in the mitotic cells of *Allium cepa* *.

Treatment	Total Cells	Mitotic Cells	Damaged Cells	Genotoxicity Percentage	Adjusted Percentage **
Negative control	215	134	12	8.9	10.9 ^a^
Sodium azide (10 ng)	195	106	87	82.1	100 ^d^
JSSE 0.1 mg	223	125	19	15.2	18.5 ^b^
JSSE 0.5 mg	208	119	21	17.6	21.5 ^b^
JSSE 1 mg	197	109	16	14.7	17.9 ^b^
JSSE 5 mg	200	120	29	24.2	29.5 ^c^

* The values represent the average of three repetitions. ** Mean values with different letters (a, b, c, and d) indicate differences (*p* < 0.05) between samples, as compared with the sodium azide control.

**Table 3 foods-08-00693-t003:** Degree of acceptance * of different pâté treatments by consumers.

Treatment **	Attribute							
	Color	Odor	Tuna Flavor	Saltiness	Spreadability	Cohesiveness	Aftertaste	Global
Control	3.2 ^b^	4.7 ^b,c^	4.8 ^b^	5.1 ^a^	4.1 ^b^	4.8 ^a,b^	5.0 ^b^	4.4 ^a^
BHA	3.4 ^b^	4.3 ^c^	4.6 ^b^	5.3 ^a^	4.0 ^b^	4.5 ^b^	4.6 ^b^	4.0 ^a^
P1	5.2 ^a^	5.8 ^a,b^	5.9 ^a^	5.7 ^a^	5.3 ^a^	5.5 ^a,b^	6.0 ^a^	5.6 ^a^
P2	6.1 ^a^	5.7 ^a,b^	6.0 ^a^	5.8 ^a^	5.2 ^a^	5.6 ^a^	6.2 ^a^	5.8 ^a^

* The values represent the average of the results obtained from 70 panelists. Mean values with different letters (a, b, and c) are significantly different (*p* < 0.05) due to the addition of the jumbo squid skin extract. ** The BHA abbreviation denotes butylated hydroxyanisole, and P1 and P2 denote the concentration of jumbo squid skin extracts corresponding to 0.05% and 0.1%, respectively, according to the Materials and Methods section.

**Table 4 foods-08-00693-t004:** Effect of the addition of the jumbo squid skin extract on the *L*, *a*, and *b* color parameters *.

Treatment **	*L**	*a**	*b**
Control	75.4 ^a^	−0.2 ^c^	12.4 ^b^
BHA	70.6 ^b^	−0.5 ^c^	13.7 ^a^
P1	65.5 ^d^	1.7 ^b^	12.9 ^a,b^
P2	67.2 ^c^	2.8 ^a^	13.6 ^a^

* Mean values are of three replicates (*n* = 3). Values with different letters (a, b, c, and d) indicate significant differences (*p* < 0.05) among the treatments. ** Abbreviations of BHA and jumbo squid extracts (P1 and P2) are the same as those expressed in Table 3.

**Table 5 foods-08-00693-t005:** Effect of the addition of the jumbo squid skin extract on the color difference (ΔE*_ab_) of the different processed pâté samples at 4 and 8 °C *.

Treatment **	4 °C						8 °C					
				Day						Day		
	0	4	8	12	16	20	0	4	12	8	16	20
Control	0	5.3 ^a,D^	6.5 ^a,C^	7.3 ^a,C^	8.9 ^b,B^	16.2 ^a,A^	0	7.6 ^a,D^	7.9 ^a,D^	8.7 ^a,C^	11.1 ^a,B^	15.3 ^a,A^
BHA	0	3.0 ^b,C^	3.2 ^b,C^	5.6 ^b,B^	9.9 ^a,A^	10.1 ^b,A^	0	0.9 ^c,E^	1.6 ^c,D^	2.5 ^b,C^	6.9 ^c,B^	8.2 ^c,A^
P1	0	1.2 ^c,C^	3.0 ^b,B^	3.9 ^c,B^	6.9 ^c,A^	7.2 ^d,A^	0	2.7 ^b,D^	2.8 ^b,D^	3.6 ^c,C^	6.2 ^c,B^	8.1 ^c,A^
P2	0	2.5 ^b,C^	3.4 ^b,B^	3.6 ^c,B^	8.7 ^b,A^	9.1 ^c,A^	0	1.0 ^c,D^	1.7 ^c,D^	3.1 ^c,C^	7.8 ^b,B^	9.3 ^b,A^

* Mean values are of three replicates (*n* = 3). Values with different lowercase letters (a, b, c, and d) indicate significant differences (*p* < 0.05) among the treatments, and values with different uppercase letters (A, B, C, D, and E) indicate significant differences during storage time. ** Abbreviations of BHA and jumbo squid extracts (P1 and P2) are the same as those shown in Table 3.

**Table 6 foods-08-00693-t006:** Total microbial count (CFU g^−1^) of mesophiles and psychrophiles * in the pâté samples at 4 and 8 °C.

	Mesophiles (4 °C)						Mesophiles (8 °C)					
	Day						Day					
**Treatment ****	**0**	**4**	**8**	**12**	**16**	**20**	**0**	**4**	**8**	**12**	**16**	**20**
Control	BDL *	10	258	3617	8365	8700	BDL	26	68	560	5400	UNC **
BHA	BDL	57	100	570	6050	8500	BDL	25	32	120	1200	UNC
P1	BDL	10	14	17	28	35	BDL	3	4	6	10	310
P2	BDL	3	4	6	7	10	BDL	2	3	5	20	190
	**Psychrophiles (4 °C)**						**Psychrophiles (8 °C)**					
	**Day**						**Day**					
**Treatment ****	**0**	**4**	**8**	**12**	**16**	**20**	**0**	**4**	**8**	**12**	**16**	**20**
Control	BDL	540	910	UNC	UNC	UNC	BDL	620	1230	7600	UNC	UNC
BHA	BDL	30	100	1500	UNC	UNC	BDL	760	1020	1580	UNC	UNC
P1	BDL	10	120	1200	2500	UNC	BDL	240	310	3100	9700	UNC
P2	BDL	10	110	1000	2900	UNC	BDL	190	280	3000	8100	UNC

* Mean value of three replicates (*n* = 3). ** Abbreviations of BHA and concentrations of jumbo squid extracts (P1, P2) are the same as those expressed in Table 3. BDL = below detection limits. UNC = the value could not be determined due to an excessive number of colonies.

**Table 7 foods-08-00693-t007:** Conjugated dienes and trienes formation * in pâté samples at 4 and 8 °C.

Conjugated Dienes (μmol g^−1^)										
**Treatment ****	**4 °C**					**8 °C**				
	**Day**					**Day**				
	**4**	**8**	**12**	**16**	**20**	**4**	**8**	**12**	**16**	**20**
Control	1.8 ^a,C^	1.1 ^b,C^	1.9 ^a,C^	3.7 ^a,B^	5.2 ^a,A^	2.8 ^a,B^	2.3 ^a,C^	2.7 ^a,B^	2.0 ^a,C^	4.9 ^b,A^
BHA	2.3 ^a,A^	0.5 ^b,C^	1.7 ^a,B^	2.1 ^b,A^	2.3 ^b,A^	1.5 ^a,C^	2.3 ^b,B^	1.6 ^b,C^	2.1 ^c,B^	3.4 ^b,A^
P1	2.8 ^a,A^	1.0 ^b,D^	1.7 ^a,C^	2.8 ^a,b,A^	2.2 ^b,B^	2.0 ^a,C^	1.3 ^b,D^	4.1 ^a,A^	4.3 ^b,A^	2.8 ^b,B^
P2	3.3 ^a,B^	2.0 ^a,C^	0.6 ^b,D^	2.3 ^b,C^	5.3 ^a,A^	2.3 ^a,C^	1.7 ^b,D^	2.2 ^b,C^	5.3 ^b,B^	9.6 ^a,A^
**Conjugated trienes** **(μmol g^−1^)**										
**Treatment ****	**4 °C**					**8 °C**				
	**Day**					**Day**				
	**4**	**8**	**12**	**16**	**20**	**4**	**8**	**12**	**16**	**20**
Control	0.5 ^a,C^	0.6 ^b,C^	1.3 ^a,B^	2.5 ^b,A^	2.1 ^b,A^	1.8 ^a,C^	2.4 ^a,B^	1.5 ^b,C^	3.8 ^a,A^	3.6 ^b,A^
BHA	0.5 ^a,D^	0.6 ^b,D^	1.2 ^a,C^	1.6 ^c,B^	3.1 ^a,A^	0.9 ^b,D^	1.6 ^b,C^	1.9 ^a,b,B^	1.5 ^b,C^	3.3 ^b,A^
P1	1.0 ^a,C^	0.7 ^b,D^	1.5 ^a,B^	1.9 ^c,A^	2.2 ^b,A^	1.3 ^b,C^	1.0 ^b,C^	2.6 ^a,B^	3.5 ^a,A^	2.5 ^c,B^
P2	0.9 ^a,C^	1.7 ^a,B^	0.8 ^a,C^	3.4 ^a,A^	3.7 ^a,A^	2.2 ^a,C^	1.2 ^b,D^	1.9 ^a,b,C^	4.2 ^a,B^	7.6 ^a,A^

* Mean values are of three replicates (*n* = 3). Values with different lowercase letters (a, b, c, and d) indicate significant differences (*p* < 0.05) among the treatments, and values with different uppercase letters (A, B, C and D) indicate significant differences during storage time. ** Abbreviations of BHA and jumbo squid pigmented extracts (P1 and P2) are the same as those shown in Table 3.

**Table 8 foods-08-00693-t008:** Sensory quality ^a^ evaluation during the storage of pâté samples ^b^ at 4 and 8 °C.

Storage Conditions		Treatment			
Temperature	Day	Control	BHA	P1	P2
4 °C	0	E			
	4	E	E	E	E
	8	B	C	A	B
	12	B	C	B	C
	16	D	D	C	D
	20	D	D	C	D
8 °C	0	E			
	4	E	E	E	E
	8	B	C	A	B
	12	B	C	A	B
	16	D	D	C	D
	20	D	D	C	D

^a^ Quality codification: excellent (E), good (A), fair (B), poor (C), and unacceptable (D). ^b^ Abbreviations of BHA and concentrations of jumbo squid extracts (P1, P2) are the same as those expressed in Table 3.

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
