# Peer review of "Evaluation of Dosidicus gigas Skin Extract as An Antioxidant and Preservative in Tuna Pâté"

_foods, 2019, doi:10.3390/foods8120693_

Round 1

Reviewer 1 Report

The article entitled "Evaluation of Dosidicus gigas Skin Extract as an Antioxidant and Preservative in Tuna Pâté" is very interesting and generally correctly written. The materials and methods section is well presented as well as the results and discussion.

However, I am confused with the statistical analysis used by the author(s) to support their findings.

1. In paragraph "2.4. Statistical analysis" the author(s) say " The data obtained from the consumer acceptance study were evaluated through a two-way analysis of variance (ANOVA) in order to assess the significance of the JSSE addition on all of the studied characteristics". I do not think that 2-way ANOVA was employed. Based on Table 3, what the reader sees is 1-way ANOVA for every sensory attribute with 4 levels, namely, P1, P2, Control and BHA.

2. In Tables 5 and 7 a 2-way ANOVA may be employed, since there are two factors, namely, storage time (0-20 days - 6 levels) and treatment (P1, P2, control and BHA - 4 levels). Storage temperature may also be considered an additional factor (4 and 8 oC - 2 levels).

3. In lines 202-203 the author(s) say "A comparison of the means was performed using the Tukey test (p < 0.05)". It should be Tukey's honestly significant difference test.

Author Response

Thank you for your suggestions and comments.  In the main document, the substantive amendments are indicated in blue. We respond to each comment in the following paragraphs:

Comments to Authors:

The article entitled "Evaluation of Dosidicus gigas Skin Extract as an Antioxidant and Preservative in Tuna Pâté" is very interesting and generally correctly written. The materials and methods section is well presented as well as the results and discussion.

ANSWER: We, the authors, are very glad you appreciated our work. Your comments and suggestions mean a lot to us.

However, I am confused with the statistical analysis used by the author(s) to support their findings.

ANSWER: Thank you for your comments. We apologize for the inconvenience. We tried our best to clarify this information in the paragraphs that you indicated.

In paragraph "2.4. Statistical analysis" the author(s) say " The data obtained from the consumer acceptance study were evaluated through a two-way analysis of variance (ANOVA) in order to assess the significance of the JSSE addition on all of the studied characteristics". I do not think that 2-way ANOVA was employed. Based on Table 3, what the reader sees is 1-way ANOVA for every sensory attribute with 4 levels, namely, P1, P2, Control and BHA.

ANSWER: Thank you for pointing this out. You are correct and we apologize for this mistake. It has been corrected in the text.

Lines 198-200, page 5: The data obtained from the consumer acceptance study were evaluated through a one-way analysis of variance (ANOVA) in order to assess the significance of the JSSE addition on all of the studied characteristics. Descriptive statistics was used to present the antioxidant assays (n = 3). The data obtained from the consumer acceptance study were evaluated through a one-way analysis of variance (ANOVA) in order to assess the significance of the JSSE addition on all of the studied characteristics. Data from color and lipid oxidation of pâté samples was evaluated thorugh a two-way analysis of variance (ANOVA).

In Tables 5 and 7 a 2-way ANOVA may be employed, since there are two factors, namely, storage time (0-20 days - 6 levels) and treatment (P1, P2, control and BHA - 4 levels). Storage temperature may also be considered an additional factor (4 and 8 oC - 2 levels).

ANSWER: Thank you for your comments. Indeed, the two-way ANOVA was now used in color and lipid oxidation of the pâté samples. We followed your advice and reanalyzed our data. Tables 5 and 7 have been modified, as well as Material and Methods      .

Lines 198-202 (page 5): Descriptive statistics was used to present the antioxidant assays (n = 3). The data obtained from the consumer acceptance study were evaluated through a one-way analysis of variance (ANOVA) in order to assess the significance of the JSSE addition on all of the studied characteristics. Data from color and lipid oxidation of pâté samples was evaluated through a two-way analysis of variance (ANOVA).

In lines 202-203 the author(s) say "A comparison of the means was performed using the Tukey test (p < 0.05)". It should be Tukey's honestly significant difference test.

ANSWER: Thank you for your observation. It has been now corrected in the text:

Lines 203-204 (page 5): A comparison of the means was performed using the Tukey’s honestly significant test (p < 0.05).

Reviewer 2 Report

The submitted manuscript gives valuable results and it might be helpful for the researchers in the field of study. These results provide a biological beneficial source of functional compounds using jumbo squid skin by-product. Findings in the submitted manuscript provide valuable information, however the author is required to confirm all the submitted manuscript by referring to the comments below;

[Abstract]

Line 15: Squid skin extract or squid skin pigmented extract were indicated in all manuscript. It is confusing to distinguish whether it is extract or pigment.

Line 20: Microbiological à microbiological analysis

Line 22: Provide full names of BHA.

Line 28: Potential what and why, and what kind of compound can it affect for the antioxidant and antimicrobial activities? (The author might have assumed that JSSE could be a potential source)

[Materials and Methods]

Line 80: The author used methanol for extraction of JSSE. As we know that methanol is a kind of toxic, therefore a little residue of methanol could be a serious problem in food product. However, the manuscript didn’t provide the results of methanol analysis. If possible, it is required to show the results.

Line 110: Is sodium azide (10ng) powder or liquid? I guess this could be a solution which has 10 ng of sodium azide.

Line 125: Butyl hydroxy anisole (0.1 g % of pâté) à Butyl hydroxy anisole (0.1 % of pâté); delete ‘g’.

Line 160: The author indicated log CFU g-1 for total plate count. However, in Table 6, it showed that it doesn’t have unit (log CFU g-1). Additionally, in Table 6, the figures ‘log number’ (ex, 8400 CFU/g à 3.93 log CFU/g) was not indicated.

Line 180: The author indicated malondialdehyde equivalents for lipid oxidation. However, it was not shown in tables or figures.

The resolution of all the figures is too blurry to understand. It is required to enhance the resolution of all the figures in the submitted manuscript.

Table 2: S.D was not presented in all figures (see the footnote in Table 2; *The values represent the average of three repetitions ± standard deviation)

Table 6: There is no unit such as log CFU/g.

Table 7: Treatment order (control, P1, P2, BHA) is different from comparing other tables (P1,P2, Control, BHA). It is required to unify the treatment order for better understanding.

Table 8: It is needed to check the comparison of 12 days between 4 ℃ and 8 ℃ of P1 and P2.

Author Response

Thank you for your suggestions and comments.  In the main document, the substantive amendments are indicated in blue. We respond to each comment in the following paragraphs:

Comments to Authors:

The submitted manuscript gives valuable results and it might be helpful for the researchers in the field of study. These results provide a biological beneficial source of functional compounds using jumbo squid skin by-product. Findings in the submitted manuscript provide valuable information, however the author is required to confirm all the submitted manuscript by referring to the comments below;

ANSWER: We, the authors, are very glad you appreciated our work. Your comments and suggestions mean a lot to us.

Line 15: Squid skin extract or squid skin pigmented extract were indicated in all manuscript. It is confusing to distinguish whether it is extract or pigment.

ANSWER: Thank you for pointing this out. The true nature of the employed natural additive is an extract. It has been now corrected throughout the manuscript.

Line 77 (page 2): and extraction temperature on the recovery of extracts with antioxidant activity Line 79 (page 2): recovery of extracts with antioxidant activity. Line 209 (page 6): 1. Preliminary analysis of jumbo squid skin extracts Line 214 (page 6): the extracts did not double the number of spontaneous revertant colonies Lines 251-252 (page 7): Abbreviations of BHA and jumbo squid extracts (P1 and P2) are the same Lines 257-258 (page 7): Abbreviations of BHA and jumbo squid extracts (P1 and P2) are the same Line 265, Table 6 (page 8): Abbreviations of BHA and jumbo squid extracts (P1 and P2) are the same Line 271, Table 7 (page 8): Abbreviations of BHA and jumbo squid extracts (P1 and P2) are the same Line 285 (page 9): Abbreviations of BHA and jumbo squid extracts (P1 and P2) are the same Line 290 (page 9): Abbreviations of BHA and jumbo squid extracts (P1 and P2) are the same Line 309-310 (page 10): Abbreviations of BHA and jumbo squid extracts (P1 and P2) are the same Line 315 (page 10): … of the extracts was measured in terms of their DPPH* and ABTS•* radical… Line 322 (page 10): mutagenicity of the extracts was considered negative Line 402 (page 13): This work demonstrated that pigments extracted from squid skin pigments extracts, with antioxidant and antimicrobial activity, as well as no mutagenic nor clastogenic effect

Line 20: Microbiological à microbiological analysis

ANSWER: Thank you for your correction. It has been now changed in the text.

Line 20 (page 1): …the water activity (aw), pH, color (Eab*), microbiological analysis, lipid oxidation…

Line 22: Provide full names of BHA.

ANSWER: Thank you for mentioning this. We now have put the full name of BHA.

Lines 21-22 (page 1): … and the results were compared with the results of butylated hydroxyanisole (BHA) and control treatments.

Line 28: Potential what and why, and what kind of compound can it affect for the antioxidant and antimicrobial activities? (The author might have assumed that JSSE could be a potential source)

ANSWER: Thank you for your observations. We have modified the manuscript, in order to follow your indications.

Lines 28-30 (page 1): This study showed that JSSE has potential biologically active pigments that can act as an antioxidant and antimicrobial in yellowfin tuna fish pâtés.

Line 80: The author used methanol for extraction of JSSE. As we know that methanol is a kind of toxic, therefore a little residue of methanol could be a serious problem in food product. However, the manuscript didn’t provide the results of methanol analysis. If possible, it is required to show the results.

ANSWER: Thank you for your comments. We understand your concerns about the possible toxicity of the methanolic squid skin extracts. While we do not present results about the amount of residual methanol in the samples, based on established limits (we kindly ask the reviewer to check lines 324 to 330), the methanolic extracts showed null mutagenic nor clastogenic effect. Moreover, the squid skin extract was used after methanol was evaporated and dissolved in vinegar for the pâté inclusion.

Line 110: Is sodium azide (10ng) powder or liquid? I guess this could be a solution which has 10 ng of sodium azide.

ANSWER: Thank you for indicating this mistake. It has been now modified.

Lines 109-110 (page 3): A positive control with a sodium azide solution (10 ng mL-1) and a negative control (water) were also analyzed.

Line 125: Butyl hydroxy anisole (0.1 g % of pâté) à Butyl hydroxy anisole (0.1 % of pâté); delete ‘g’.

ANSWER: We apologize for this error. It has been now corrected.

Line 125 (page 3): Butyl hydroxy anisole (0.1 % of pâté) was added to a third…

Line 160: The author indicated log CFU g-1 for total plate count. However, in Table 6, it showed that it doesn’t have unit (log CFU g-1). Additionally, in Table 6, the figures ‘log number’ (ex, 8400 CFU/g à 3.93 log CFU/g) was not indicated.

ANSWER: We thank the reviewer for the observation. The following information has been modified:

Lines 159-160 (page 4): The average count was multiplied by the dilution factor and expressed as the CFU g-1 of the sample.

Line 180: The author indicated malondialdehyde equivalents for lipid oxidation. However, it was not shown in tables or figures.

ANSWER: Thank you for pointing this out. It has been now corrected in the main manuscript.

Line 180 (page 4): … and the results were expressed as mg of malondialdehyde per kg of pâté.

The resolution of all the figures is too blurry to understand. It is required to enhance the resolution of all the figures in the submitted manuscript.

ANSWER: We apologize for this major inconvenience. The figures have been now modified.

Table 2: S.D was not presented in all figures (see the footnote in Table 2; *The values represent the average of three repetitions ± standard deviation)

ANSWER: Thank you for your observation. The information has been modified.

Line 224, Table 2 (page 6): The total cells column has been changed to express the average of cells counted. Line 224, Table 2 (page 6): *The values represent the average of three repetitions

Table 6: There is no unit such as log CFU/g.

ANSWER: Thank you for pointing this out. This has been corrected in the manuscript. The units that will be used are CFU g-1.

Lines 159-160 (page 4): The average count was multiplied by the dilution factor and expressed as the CFU g-1 of the sample. Line 265 (page 8): Table 6. Total microbial count (CFU g-1) of mesophiles and psychrophiles* in the pâté samples at 4 and 8 °C. Lines 354-356 (page 12): The maximum allowed levels of microorganisms established for meat products, cured and cooked meat products, and emulsified and cooked cured products are 250 log CFU g-1 [29]. Species-specific spoilage psychrotrophs bacteria,

Table 7: Treatment order (control, P1, P2, BHA) is different from comparing other tables (P1,P2, Control, BHA). It is required to unify the treatment order for better understanding.

ANSWER: The authors really appreciate your observation. Tables have been now modified in order to follow the same order of the treatments.

Line 231, Table 3 (page 7): The order is Control, BHA, P1, P2 Line 249, Table 4 (page 7): The order is Control, BHA, P1, P2 Line 253, Table 5 (page 7): The order is Control, BHA, P1, P2

Table 8: It is needed to check the comparison of 12 days between 4 ℃ and 8 ℃ of P1 and P2.

ANSWER: Thank you for your observations. We have added new information on the matter in the following lines.

Lines 388-391 (page 12): The previous results were unexpected because the chemical and microbiological assays suggested that the P2 sample (0.1 % of JSSE) had better results than the other treatments. P2 had the same overall sensory scores compared with control, whereas BHA had lower scores throughout the experiment. This could be attributed to the development and increased production of tuna odors and those associated to the addition of BHA and a higher squid skin extracts during storage time.

Reviewer 3 Report

The paper discusses a method for food preservation, based on a methanol-HCl squid skin extract (Dosidicus 15 gigas) (JSSE) by determining its antioxidant and mutagenic activities. It shows that JSSE is a potent food preserver.

Th experiments are discussed thoroughly and the article is written very well overall. There are some minor typo errors due to word processing software, I believe (line 70, no space after period).

Author Response

Thank you for your suggestions and comments.  In the main document, the substantive amendments are indicated in blue. We respond to each comment in the following paragraphs:

Comments to Authors:

The paper discusses a method for food preservation, based on a methanol-HCl squid skin extract (Dosidicus gigas) (JSSE) by determining its antioxidant and mutagenic activities. It shows that JSSE is a potent food preserver.

ANSWER: We, the authors, are very glad you appreciated our work. Your comments and suggestions mean a lot to us. We agree with the reviewer. Squid skin extracts had a very potent activity as an antioxidant and as an antimicrobial agent.

Th experiments are discussed thoroughly and the article is written very well overall. There are some minor typo errors due to word processing software, I believe (line 70, no space after period).

ANSWER: Thank you for pointing this out. It has been corrected.

Line 70. The space was added before the period.

Round 2

Reviewer 2 Report

The article is well-revised and organized following the reviewer’s suggestions.

Additionally, the responses to comments are appropriate, and there are not any further comments.